# Neuroticism and Stress in Older Adults: The Buffering Role of Self-Esteem

**DOI:** 10.3390/ijerph20126102

**Published:** 2023-06-12

**Authors:** Maya E. Amestoy, Danielle D’Amico, Alexandra J. Fiocco

**Affiliations:** 1Department of Psychological Clinical Science, University of Toronto Scarborough, 1265 Military Trail, Toronto, ON M1C 1A4, Canada; maya.amestoy@mail.utoronto.ca; 2Department of Psychology, Toronto Metropolitan University, 350 Victoria Street, Toronto, ON M5B 2K3, Canada

**Keywords:** neuroticism, perceived stress, coping, self-esteem, older adults

## Abstract

Background: Chronic stress is associated with accelerated aging and poor health outcomes in older adults. According to the Transactional Model of Stress (TMS), distress is experienced when one perceives the stressor, or threat, to outweigh the ability to cope. The experience of distress is correlated with trait neuroticism, which is associated with greater perceptions of stress and stress reactivity, as well as a tendency to engage in maladaptive coping strategies. However, as individual personality traits do not act in isolation, this study aimed to investigate the moderating role of self-esteem in the relationship between neuroticism and distress using a TMS framework. Methods: A total of 201 healthy older adults (Mage = 68.65 years) completed questionnaires measuring self-esteem, neuroticism, perceived stress, and positive coping. Results: Greater neuroticism was significantly associated with less positive coping at low (b = −0.02, *p* < 0.001) and mean self-esteem levels (b = −0.01, *p* < 0.001), but not at high self-esteem levels (b = −0.01, *p* = 0.06). No moderating effect was found for perceived stress or overall distress. Conclusion: The results support the association between trait neuroticism and indices of stress and suggest a potential buffering effect of self-esteem in moderating the negative association between neuroticism and positive coping.

## 1. Introduction

Chronic stress or frequent intermittent stress plays a key role in the development and maintenance of a range of physical and psychological health ailments across the lifespan, with increased vulnerability later in life [1,2]. Chronic activation of stress-sensitive systems has been implicated in various disorders among older adults, including insomnia [3], late-life depression [4], anxiety [5], cognitive impairment, and Alzheimer’s disease [6].

Research demonstrates substantial individual differences in the perception of stress and the ability to manage stress when it arises [7]. According to the Transactional Model of Stress (TMS) [8], a three-process cognitive model, the experience of stress is dependent on the interplay between an individual and their environment, which is appraised as taxing or exceeding their resources to adapt. Primary appraisal entails evaluating whether the environment or stimulus is irrelevant, benign-positive, or stressful. If the environment is appraised as harmful, threatening, or challenging, secondary appraisal ensues, which involves evaluating the resources or coping strategies that are available in order to adapt to the stressor. Distress arises when the perceived stressor outweighs available internal and external resources, that is, when coping resources are perceived as inadequate. Alternatively, if coping resources outweigh the taxing situation, distress is minimized. One key factor that contributes to individual differences in stress appraisal is trait neuroticism [9].

Neuroticism is a personality trait that encompasses an increased tendency to experience negative affective states, namely, anger, anxiety, irritability, and depression [10]. High trait neuroticism is a well-known risk factor for the development of psychopathology [11] and plays a key role in the etiology and course of various psychological disorders across the lifespan, including later adulthood [12]. Indeed, the association between high trait neuroticism and poorer psychological well-being has been shown to be stronger among middle-aged and older adults compared with younger adults [13]. Early research suggests that individuals high in trait neuroticism are more likely to appraise ambiguous situations as negative or threatening compared to their low trait neuroticism counterparts [14]. Compared to lower trait neuroticism, higher trait neuroticism is associated with higher baseline stress perception [15] and heightened reports of perceived threat [16,17].

In addition to greater threat perception, persons high in trait neuroticism also display deficits in the ability to cope with stressors. Specifically, higher trait neuroticism is associated with greater use of maladaptive coping strategies [15,18] such as disengagement, self-blame, and rumination [19], and lower use of adaptive coping strategies including problem solving or task-focused coping [20]. Consequently, according to the TMS, persons with high trait neuroticism may be more prone to stress and stress-related ailments due to an imbalance between the perceived threat and the ability to cope with that threat [9].

Although robust associations exist between neuroticism and perceptions of stress, it is important to acknowledge that individual personality traits are presented along a continuum and do not act in isolation. For example, a longitudinal analysis of substance use among participants enrolled in the Midlife in the United States (MIDUS) Study revealed that the positive association between trait neuroticism and alcohol consumption was moderated by trait conscientiousness, such that higher trait neuroticism was less predictive of greater alcohol consumption among persons high in trait conscientiousness [21]. As such, traits that are protective may minimize the impact of trait neuroticism on perceptions of stress and the ability to cope.

Self-esteem is a personality construct defined as the global assessment of one’s positive or negative attitude toward oneself [22]. Research suggests that lower self-esteem is associated with greater perceived stress and stress reactivity [23], whereas higher self-esteem is associated with lower reactivity to stressful events [24]. Brown [25] showed that high self-esteem individuals were more resilient and less distressed than those with low self-esteem when faced with negative social feedback following an interpersonal encounter. Furthermore, persons who score high on self-esteem are more likely to display adaptive coping strategies, such as problem solving, planning, and positive re-interpretation, whereas low self-esteem individuals tend to employ maladaptive strategies, such as denial and behavioral disengagement [26].

Extant research has demonstrated an inverse relationship between neuroticism and self-esteem [27]. When neuroticism levels are high, self-esteem may be minimized due to feelings of self-doubt, self-consciousness, and guilt [28]. Research also shows that neuroticism and self-esteem may interact to increase the risk of depressive symptoms in young and middle-aged adults [28], supporting the need to evaluate these two personality constructs in parallel when evaluating stress and stress-related outcomes.

Building on the extant literature, the objective of the current study was to examine the interplay between neuroticism, self-esteem, and perceived stress among community-dwelling older adults. Employing the TMS as a framework, it may be surmised that older adults with high trait neuroticism may experience greater stress on a more frequent basis as they are more likely to perceive stressors in their environment and are less likely to possess the resources necessary to cope with those stressors. However, self-esteem may moderate this association by minimizing the perception of depleted resources, thus weakening the association between neuroticism and perceived stress. Accordingly, this study examined whether self-esteem moderates the association between neuroticism and (1) self-reported level of distress; (2) the extent to which the environment is perceived as stressful or helpless; and (3) the perceived availability of resources to adapt to the perceived stress. It was hypothesized that self-esteem would buffer the relationship between higher trait neuroticism and greater overall perceived stress, perceived distress or helplessness, and the perceived ability to engage in adaptive coping strategies.

## 2. Materials and Methods

### 2.1. Participants

Community-dwelling older adults (*n* = 201) aged 60 and older were recruited as part of a larger study through community board messages, online advertisements, and the university participant pool. Study participation included two laboratory visits, one of which involved completing a battery of psychosocial questionnaires. Exclusion criteria included learning English after the age of 12, presenting with uncorrected vision or hearing impairment, and having a diagnosis of a neurological disorder (e.g., Parkinson’s disease), substance abuse/dependency, schizophrenia spectrum disorder, bipolar disorder, or a learning disability. Participants were also excluded if they reported a history of chemotherapy or radiation treatment, head injury, or underwent general anesthesia in the last year. Exclusion criteria were chosen due to their known influence on cognitive performance, the primary outcome of the larger study. With a mean age of 68.65 years (SD = 6.95), 63.2% of the sample were female and 85.1% were Caucasian. Overall, the sample was well educated (16.13 years (SD = 4.25) of education), with 69.5% identifying as having a medium socioeconomic status (SES). All participants consented to participate in the study, which was approved by Toronto Metropolitan University’s Research Ethics Board (REB 2014-164). A summary of participant characteristics is shown in Table 1.

### 2.2. Measures

#### 2.2.1. Sociodemographic Information

Sociodemographic questions were completed to index age, biological sex, ethnicity, years of education, and perceived socioeconomic status (SES; low, middle, high).

#### 2.2.2. Perceived Stress and Coping

The 10-item Perceived Stress Scale (PSS-10) [29] measures how often the participant felt that their life was unpredictable, uncontrollable, and overloaded in the last 12 months, on a five-point Likert scale (0 = never to 4 = very often). The scale comprises 6 negatively worded items to index stressful thoughts and feelings, and 4 positively worded items to index self-efficacy or positive coping. Higher scores (range 0–40) suggest greater perceived distress. For the purpose of the current study, total PSS-10 and its two sub-scores were used to operationalize the current study outcomes. Past research supports a 2-factor structure of the PSS-10, including perceived helplessness or stress, and perceived self-efficacy or positive coping [30,31], which aligns with the TMS interplay between perceived threat and perceived resources to cope with the threat [8]. The PSS-10 showed strong reliability in the current sample, including Cronbach’s α = 0.88 for the total score, α = 0.85 for the stress subscale score, and α = 0.77 for the positive coping subscale score; hereafter, referred to as total distress, perceived stress, and positive coping, respectively.

#### 2.2.3. Self-Esteem

The 10-item Rosenberg Self-Esteem Scale (RSES) [22] was used to measure the moderating variable, self-esteem. Participants reported the extent to which they agreed with each statement on a four-point Likert scale (1 = strongly agree to 4 = strongly disagree). Higher scores (range 10 to 30) indicate greater self-esteem. The RSES demonstrated good internal consistency in the current sample, with a Cronbach’s α of 0.83.

#### 2.2.4. Neuroticism

The 12-item Neuroticism subscale of the NEO Five-Factor Inventory [32] was used to assess the predictor variable, trait neuroticism. Participants reported the extent to which they identified with core characteristics of trait neuroticism using a 5-point Likert (0 = strongly disagree to 4 = strongly agree). Higher scores denote greater trait neuroticism. The neuroticism subscale demonstrated good internal consistency in the current sample, with a Cronbach’s α of 0.89.

### 2.3. Statistical Analyses

Statistical analyses were conducted using IBM SPSS Statistics v21. A priori power calculation using G*Power (v3.1.9.4) for linear multiple regression, assuming a small–medium effect size (*f*^2^ = 0.05), α = 0.05, and Power of 0.95, suggested a total sample size of 159 participants in the analytical model. As such, the current sample size was deemed suitable for the proposed secondary analyses. Perceived stress scores did not meet the assumptions of normality and therefore square root transformation (SQRT) was performed. All continuous variables were mean-centered. Pearson biserial correlations were conducted to assess zero-order correlations between all variables of interest. To test for the moderating effect of self-esteem in the relationship between neuroticism and perceived stress, SPSS PROCESS Macro (Model 1) [33] was used with 5000 bootstrapped samples. In all moderation models, a priori covariates included sex and age due to their potential associations with perceived stress, coping, and self-esteem [34]. The adjusted model also included self-reported SES as it was significantly associated with total distress (*r* = −0.25, *p* < 0.001), self-esteem (*r* = 0.30, *p* < 0.001), and trait neuroticism (*r* = −0.22, *p* = 0.002).

In all moderation models, neuroticism was entered as the independent variable, and self-esteem was entered as the moderating variable. To address the study objectives, total distress, perceived stress, and positive coping were each entered as the dependent variable in separate models. Simple slopes were estimated to determine the conditional association between neuroticism and outcome scores at low (1 SD below the mean), mean (Mean), and high levels (1 SD above the mean) of self-esteem. See Figure 1 for conceptual moderation models.

## 3. Results

### 3.1. Participant Characteristics

On average, the sample reported relatively low total perceived distress (M = 12.05, SD = 6.46; 0–13 is considered low stress) [29,35], which was also reflected in subscales scores of perceived stress (M = 5.04, SD = 2.93) and positive coping (9.08, SD = 2.08). Mean neuroticism was relatively high (M = 45.72, SD = 10.91; >33 is indicative of high neuroticism) [32], as was mean self-esteem (M = 23.20, SD = 4.41; >15 is indicative of high self-esteem) [22].

### 3.2. Bivariate Correlations

Table 2 displays the Pearson bivariate correlations between all variables entered in the analytical models. Greater trait neuroticism was significantly associated with younger age (r = −0.15, *p* = 0.04), reporting a lower perceived SES (r = −0.22, *p* = 0.002), greater total distress (r = 0.64, *p* < 0.001), lower positive coping (r = −0.62, *p* < 0.001), greater perceived stress (r = 0.54, *p* < 0.001), and lower self-esteem (r = −0.64, *p* < 0.001). Greater self-esteem was significantly associated with higher perceived SES (r = 0.30, *p* < 0.001), lower total distress (r = −0.58, *p* < 0.001), greater positive coping (r = 0.61, *p* < 0.001), and lower perceived stress (r = −0.45, *p* < 0.001). Relative to females, males reported lower total distress (r = −0.14, *p* = 0.05) and perceived stress (r = −0.15, *p* = 0.03), but did not differ on positive coping (r = 0.06, *p* = 0.37). Older age was associated with lower total distress (r = −0.15, *p* = 0.04) and perceived stress (r = −0.21, *p* = 0.003), but not positive coping (r = 0.11, *p* = 0.12).

### 3.3. Neuroticism, Self-Esteem, and Total Distress Score

The overall adjusted model was statistically significant (*F*(6, 191) = 29.50, *p* < 0. 001, R^2^ = 0.48). Analysis of main effects showed that higher trait neuroticism was significantly associated with greater self-reported total distress (b = 0.04, *t*(191) = 16.54, *p* < 0.001), whereas higher self-esteem was associated with lower total distress (b = −0.06, *t*(191) = −3.82, *p* < 0.001). No significant neuroticism-by-self-esteem interaction effect was found (b = 0.00, *t*(191) = 0.50, *p* = 0.62).

### 3.4. Neuroticism, Self-Esteem, and Perceived Stress Subscale Score

The overall adjusted model was statistically significant (*F*(6, 191) = 17.42, *p* < 0.001 R^2^ = 0.35). Analysis of main effects showed that higher trait neuroticism was significantly associated with greater perceived stress, (b = 0.02, *t*(191) = 5.39, *p* < 0.001), whereas greater self-esteem was associated with lower perceived stress (b = −0.03, *t*(191) = −2.03, *p* = 0.04). No significant neuroticism-by-self-esteem interaction effect was found (b = 0.00, *t*(191) = 0.43, *p* = 0.67).

### 3.5. Neuroticism, Self-Esteem, and Positive Coping Subscale Score

The overall adjusted model was statistically significant (*F*(6, 191) = 33.44, *p* <.001, R^2^ = 0.51). Analysis of main effects showed that higher trait neuroticism was significantly associated with lower positive coping (b = −0.25, *t*(191) = −5.09, *p* < 0.001), whereas greater self-esteem was associated with greater positive coping (b = 0.03, *t*(191) = 4.65, *p* < 0.001). A significant interaction between neuroticism and self-esteem was found (b = 0.002, *t*(191) = 4.45, *p* < 0.001). Further, the addition of the interaction term significantly improved the overall model (*F*(1, 191) = 19.80, *p* < 0.001, R^2^change = 0.05). As displayed in Figure 2, conditional process analysis suggested that greater trait neuroticism is associated with lower positive coping at the low (1 SD below the mean = −4.39, b = −0.02, *t*(191) = −7.15, *p* < 0.001, 95%CI =−0.02, −0.01) and mean (mean = 0.00, b = −0.01, *t*(191) = −5.09, *p* <.001, 95%CI= −0.02, −0.01), but not high self-esteem (one SD above the mean = 4.39, b = −0.01, *t*(191) = −1.93, *p* = 0.06, 95%CI = −0.01, 0.00).

## 4. Discussion

According to the TMS, the extent to which one experiences distress is determined by the interplay between perceived threat (i.e., the stressor) and available resources (i.e., coping strategies) to manage the perceived threat. Using the TMS as a framework, the current study examined whether self-esteem may moderate the association between neuroticism and two elements of the TMS, perceived stress and positive coping, in older adults. Although self-esteem was not found to moderate the association between neuroticism and perceived stress, or total distress, it did moderate the association between neuroticism and positive coping.

The current findings are in line with decades of research that underscore the importance of self-esteem in the uptake of adaptive coping strategies [26,36]. Research suggests that high self-esteem may offset the potential adverse effects of unpleasant experiences through positive self-worth and self-efficacy, fostering positive psychological states [37]. In a study by Lane et al. [26], higher self-esteem was associated with lower reductions in self-efficacy following failure, or defeat, among 91 national standard tennis players. Further, persons with lower self-esteem were more likely to engage in maladaptive coping strategies following failure [26]. Research also suggests that individuals with high self-esteem are able to reject negative events from their self-concept, while individuals with low self-esteem tend to internalize failure [23]. Thus, when individuals with low self-esteem encounter failure, this may reinforce negative self-perceptions and impact self-efficacy around one’s ability to engage in adaptive coping strategies. In the context of the current study, it may be surmised that older adults with greater self-esteem hold a sense of greater self-efficacy, fostering adaptive coping engagement, which may buffer the relatively robust association between trait neuroticism and maladaptive coping. Indeed, greater neuroticism was associated with lower positive coping in the low and medium self-esteem groups; however, this relationship was no longer significant in the high self-esteem group.

In contrast to the moderating role of self-esteem in the relationship between neuroticism and positive coping, self-esteem did not moderate the association between neuroticism and perceived stress, or overall distress. Null moderating effects were surprising, especially given the extant literature suggesting that self-esteem may moderate threat appraisal [38] and the stress response that ensues [39]. One plausible explanation for null findings may be that self-esteem is more robustly associated with secondary appraisals of stress (i.e., evaluating the ability to employ coping strategies to address stress) compared to primary appraisals (i.e., determining whether a stressful situation poses a threat). Somewhat aligned with this postulation are the results reported by Juth et al. [40], wherein self-esteem did not associate with the frequency of stressful experiences, despite an association between self-esteem, affect, and severity of symptoms among adults with asthma and rheumatoid arthritis. It is also possible that self-esteem and neuroticism independently contribute to perceptions of stress and overall distress. That is, self-esteem may not buffer the association between neuroticism and distress. The lack of a self-esteem buffering effect has been reported in the context of stressful life events and depression. In a longitudinal study of adolescents and young adults, low self-esteem and stressful events independently predicted subsequent depression but did not interact in predicting depression [41]. However, it is also plausible that self-esteem may only moderate overall perceptions of stress to the extent to which it is able to regulate positive coping. In the current study, effect modification in the association between neuroticism and positive coping was relatively small.

Previous research suggests that self-esteem declines in later adulthood [42]. Research also suggests that changes in self-esteem may be determined by health-related factors [43]. The current findings support the value of cultivating self-esteem to foster positive coping with life events later in life. By engaging in activities that promote self-esteem in later life, such as physical activity [44], older adults may develop a sense of self-efficacy [45] around their ability to cope with perceived stressors that arise with aging.

Interpretations of the study findings are made in light of notable study limitations. First, the cross-sectional nature of the study design precludes statements of causality pertaining to the association between neuroticism, self-esteem, and perceived stress and coping. However, this study aimed not to predict outcomes, but to examine nuances in the association between constructs, guided by the TMS framework. Second, observed effects may have been underestimated due to the sample characteristics. Self-esteem in the current sample was relatively high and mean perceived distress was relatively low, which may stem from the nature of the sample, which comprised of healthy community-dwelling older adults who were primarily Caucasian with a high education level and moderate SES. Accordingly, additional research is needed to examine the relationship between these constructs in a more diverse sample. Finally, using a TMS framework, additional research is needed to examine the interplay between neuroticism, self-esteem, stress, and coping in the prediction of well-being in later adulthood using a longitudinal research design. Future research may also address the interplay between self-esteem and other personality traits stemming from the Big Five model [10,32].

## 5. Conclusions

The current results support a buffering effect of self-esteem in moderating the relationship between neuroticism and positive coping among older adults. Although additional research is needed, these findings may be valuable for health promotion, encouraging self-esteem-promoting activities for older adults. Replication is important to further substantiate the study findings.

## Figures and Tables

**Figure 1 ijerph-20-06102-f001:**
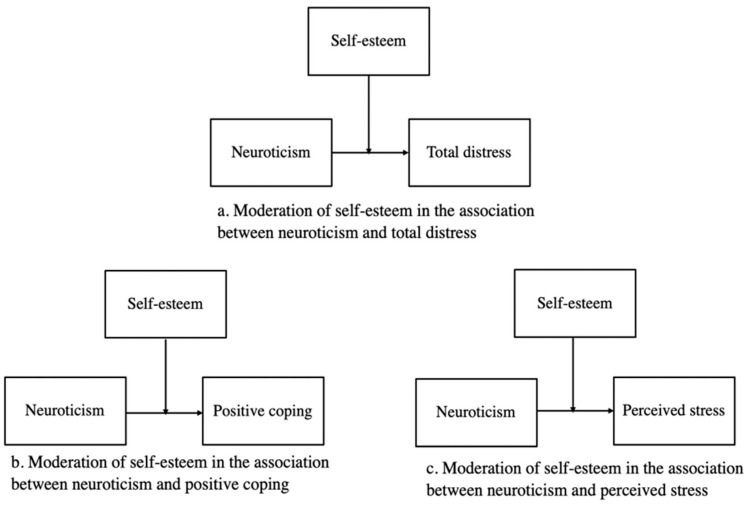
Conceptual moderation models for study outcomes total distress (**a**), perceived stress (**b**), and positive coping (**c**).

**Figure 2 ijerph-20-06102-f002:**
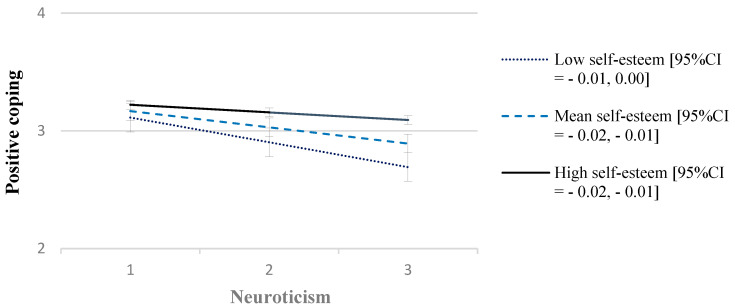
Conditional effects of the association between neuroticism and positive coping at low, mean, and high self-esteem.

**Table 1 ijerph-20-06102-t001:** Participant Characteristics.

Study Variable	Mean ± SD (Range) or % (*n*)
Sex (% female)	63.2%
Age	68.65 ± 6.95 (60–95)
Ethnicity	
Caucasian/European	85.1% (171)
Chinese	8% (4)
Black/African American	2.0% (4)
East Asian	2.5% (5)
Other	6.5% (13)
Perceived socioeconomic status	
Low	19.5% (39)
Middle	69.5% (139)
High	11.0% (22)
Years of Education	16.13 ± 4.26 (0–29)
Total Distress (raw)	12.05 ± 6.46 (0–32)
Positive Coping (raw)	9.08 ± 2.08 (0–12)
Perceived Stress (raw)	5.04 ± 2.93 (0–14)
Neuroticism	45.71 ± 10.91 (19–74)
Self-Esteem	24.20 ± 4.41 (6–30)

*Notes:* Percentage (%) reflects proportion of respondents; SD = Standard deviation.

**Table 2 ijerph-20-06102-t002:** Pearson bivariate correlations between the continuous variables of interest.

	1	2	3	4	5	6	7
Age (1)	-	-	-	-	-	-	-
Sex (2)	0.06	-	-	-	-	-	-
SES (3)	0.15 *	0.08	-	-	-	-	-
Self-esteem (4)	0.07	0.02	0.30 ***	-	-	-	-
Perceived Stress (5)	−0.15 *	−0.14 *	−0.25 ***	−0.58 ***	-	-	-
Positive Coping (6)	0.12	0.06	0.21 **	0.61 ***	−0.75 ***	-	-
Perceived Distress (7)	−0.21 **	−0.15 *	−0.21 **	−0.45 ***	0.89 ***	−0.52 ***	-
Neuroticism (8)	−0.15 *	0.02	−0.22 **	−0.64 ***	0.64 ***	−0.62 ***	0.54 ***

*Notes*. SES = socioeconomic status. * *p* < 0.05. ** *p* < 0.01. *** *p* < 0.001.

## Data Availability

Data are available on the Open Science Framework. Fiocco, A. J. (15 March 2023). Biopsychosocial Study. Retrieved from osf.io/56sa2 (accessed on 15 March 2023).

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
