# Peer review of "Neuroticism and Stress in Older Adults: The Buffering Role of Self-Esteem"

_ijerph, 2023, doi:10.3390/ijerph20126102_

Round 1

Reviewer 1 Report

The manuscript is concerned with understanding the role of self-esteem in relation to stress and neuroticism. This is a topic of interest that has already found its way into the literature. I believe the work has critical elements that I have listed below. 

Major concerns

Tthe authors should explain the choice of adopting a scale for neuroticism rather than a scale for the assessment of personality characteristics in general.

The choice of restricting the focus to neuroticism alone exposes the work to criticism of the partial role played by this factor. 

Probably, other factors contribute together with neuroticism (e.g. introversion/extroversion). Other elements could be considered, such as e.g.  self-efficacy, sutisfaction, mood (e.g. Fadda & Scalas, 2016; Costa et sal., 1981) given the paucity of the scales used. This aspect should be discussed. 

A graphic on the mederator factor could be added.

Other main concerns regard the identification of the sample size. How did the authors calculate it?  

Minor concerns

Subpara 2.2.1. Here, reference should be made to Table 1. Only the description of the sensitive variables should be included in this sub-section including some parts that are incorrectly included in the Results section.

Results relating to PSS and self-esteem should be moved to the Results section.

Author Response

Reviewer #1:

The manuscript is concerned with understanding the role of self-esteem in relation to stress and neuroticism. This is a topic of interest that has already found its way into the literature. I believe the work has critical elements that I have listed below. 

Response: We thank the reviewer for their positive overview of our manuscript. We have addressed each of their comments which are itemized below.

A few comments/suggestions/questions for the authors:

  1. The authors should explain the choice of adopting a scale for neuroticism rather than a scale for the assessment of personality characteristics in general. The choice of restricting the focus to neuroticism alone exposes the work to criticism of the partial role played by this factor. Probably, other factors contribute together with neuroticism (e.g. introversion/extroversion). Other elements could be considered, such as e.g.  self-efficacy, sutisfaction, mood (e.g. Fadda & Scalas, 2016; Costa et sal., 1981) given the paucity of the scales used. This aspect should be discussed. 

Response: We thank the reviewer for this comment. The proposed focus on neuroticism, rather than the Big Five more broadly, is largely theory driven. Within the stress literature, trait neuroticism is the only personality trait that is associated with Gray’s biopsychological theory of personality (namely, the Behavioral Inhibition System), and Sul’s Neurotic Cascade model (see citation to Suls, & Martin, 2005). A broader exploration of different personality traits and mood moves beyond the analytical framework and research question of interest. However, to address this comment, we have included the statement “Future research may also address the interplay between self-esteem and other personality traits stemming from the Big Five model [10, 32].”     

  1. A graphic on the moderator factor could be added.

Response: We thank the reviewers for this suggestion. We have now added figures depicting the models under the Statistical Analyses section, labelled as Figure 1

  1. Other main concerns regard the identification of the sample size. How did the authors calculate it?  

Response: We have included the sample size calculation in section 2.3.

4, Subpara 2.2.1. Here, reference should be made to Table 1. Only the description of the sensitive variables should be included in this sub-section including some parts that are incorrectly included in the Results section.

Response: We thank the reviewer for this suggestion. We have now removed the demographic characteristics for the results section and have added them to subparagraph 2.1.  under Materials and Methods – Participants.

  1. Results relating to PSS and self-esteem should be moved to the Results section.

Response: We are unsure as to what the reviewer is referring to here. Results related to PSS and self-esteem are in the results section.

Reviewer 2 Report

Thank you for submitting the manuscript

“Neuroticism and stress in older adults: The buffering role of 2 self-esteem”

I have some small considerations about the manuscript so that the authors can answer them.

How the measures were taken, with an online, self-administered questionnaire, who passed it...??' should reflect it in the document

When they talk about subjects in community, are they referring to those who are institutionalized? or what community are you referring to?

Line 262. Somewhat aligned with this postulation are the results reported by XXXX?? [40]

I found it an interesting study and the associations they reflect. It is true, as the authors say, studies should continue with another type of profile of the elderly.

Author Response

Reviewer 2:

I found it an interesting study and the associations they reflect. It is true, as the authors say, studies should continue with another type of profile of the elderly.

Response: We thank the reviewer for their positive overview of our manuscript. We have addressed each of their comments which are itemized below.

  1. How the measures were taken, with an online, self-administered questionnaire, who passed it...??' should reflect it in the document

Response: We thank the reviewers for this comment. We have now stated in section 2.1 “Study participation included two laboratory visits, one of which involved completing a battery of psychosocial questionnaires”

  1. When they talk about subjects in community, are they referring to those who are institutionalized? or what community are you referring to?

Response: We thank the reviewer for this question. When describing community-dwelling older adults, we are referring to individuals recruited from the general population. Using the term “community-dwelling” is common when referring to independent older adults living in their homes.

  1. Line 262. Somewhat aligned with this postulation are the results reported by XXXX?? [40]

Response: We thank the reviewer for this comment. We have now added the name of the authors “Juth et al. [40]”.

Reviewer 3 Report

The manuscript is well written and analyses are ok. I have only minor suggestions.

Abstract

Line 13: "correlated with trait neuroticism"

It should be added "personality trait"

Line 21: is it really robust?

2.3. Statistical Analyses

Line 150: "To test for the moderating effect of self-esteem [...], SPSS PROCESS Macro (Model 1) [33] was used with 5000 bootstrapped samples"

Why bootstrapping? I have nothing against it, but the authors should justify this choice since N=201 seems fine to me for running a moderation.

Table 2: it's unreadable, please reformat the table so values are aligned

Models' stats

I know it could be personal taste, but beta (instead of b) give a standardized coefficients which are easy to interpret across scales (and ranges) and give the magnitude of effects. I think it would be better providing beta coefficients (I let the authors choose)

Figure 1: I think it would be much more informative providing shaded 95%CIs with the regressed lines (eventually rescaling the Y axis)

Author Response

Reviewer #3:

The manuscript is well written and analyses are ok. I have only minor suggestions.

Response: We thank the reviewer for their feedback. We have addressed each of their comments which are itemized below.

  1. Line 13: "correlated with trait neuroticism", It should be added "personality trait"

Response: We appreciate the reviewer’s comment; however, we prefer to use the term trait neuroticism as that is the variable of interest.

  1. Line 21: is it really robust?

Response: We thank the reviewer for this comment. We have removed the word robust on line 21, the sentence now reads “Results support the association between trait neuroticism and indices of stress, and suggest a potential buffering effect of self-esteem in moderating the association between neuroticism and less positive coping.”

  1. Line 150: "To test for the moderating effect of self-esteem [...], SPSS PROCESS Macro (Model 1) [33] was used with 5000 bootstrapped samples" Why bootstrapping? I have nothing against it, but the authors should justify this choice since N=201 seems fine to me for running a moderation.

Response: We thank the reviewer for this comment. Given the assumption of normality was not met, bootstrapping was applied to the model based on the central limit theorem and in an effort to enhance the accuracy of the model estimates.

  1. Table 2: it's unreadable, please reformat the table so values are aligned

Response: We thank the reviewer for pointing this out. We have now reformatted the table.

  1. I know it could be personal taste, but beta (instead of b) give a standardized coefficients which are easy to interpret across scales (and ranges) and give the magnitude of effects. I think it would be better providing beta coefficients (I let the authors choose).

Response: We appreciate the reviewer for this suggestion. We have chosen to use b as the PROCESS model does not provide beta estimates.  

  1. Figure 1: I think it would be much more informative providing shaded 95%CIs with the regressed lines (eventually rescaling the Y axis)

Response: We thank the reviewers for this suggestion. Error bars and 95%CI are included in the revised graph.  

Round 2

Reviewer 2 Report

Thank you very much for the review, I consider that it can be published.